# Static load test and bearing capacity analysis of broken line pretensioned prestressed concrete I-beam

Peisen Wang[1,2], Jiacheng Shi[1], Chenning Song[2]*

**1** School of Civil Engineering, Shandong Jianzhu University, Jinan, China, **2** Key Laboratory of Building Structural Retrofitting and Underground Space Engineering, Shandong Jianzhu University, Ministry of Education, Jinan, China

* scanner2007@163.com

## Abstract

The construction technology of broken line pretensioned method, which is suitable for long-span bridges, can greatly improve the flexural strength of concrete members. To study the mechanical performance and failure mode of broken line pretensioned prestressed concrete I-beam, an I-beam with a span of 30 m was taken as the research object and on-site static load tests were carried out. Based on the ABAQUS finite element software, the numerical simulation was carried out to analyze the mechanical properties and failure behavior of the I-beam under the concentrated load, and the ultimate flexural capacity of the I-beam was studied in combination with the theoretical analysis. In addition, the optimized arrangement of prestressed tendons was analyzed. The results show that in the on-site static load tests, the strain values at the mid-span section and 1/4-span basically show a linear growth trend, and the I-beam is always in the elastic stage during the tests. The numerical simulation results are in general agreement with the test data with regard to strain, displacement, and neutral axis position. When the concentrated load reaches 1910 kN, the prestressed tendons at the bottom of the I-beam enter the yield state. The prestressed tendons play a controlling role in the ultimate bearing capacity of the I-beam. The numerical simulation result of the ultimate bending moment in the mid-span section is 3.55% larger than the theoretical analysis result, and the theoretical calculation formulas tend to be conservative. When the prestressed tendons are arranged by the four-fold point method, the I-beam has higher stiffness and bearing capacity after concrete cracking. The research results can provide a reference for the design and construction of broken line pretensioned prestressed concrete I-beam.

**Data availability statement:** All relevant data are within the manuscript and its Supporting Information files.

**Funding:** the Shandong Provincial Natural Science Foundation, China ( ZR2021ME238, ZR2024ME056) the National Natural Science Foundation, (51908338) the Shandong Provincial Youth Innovation Team Plan, (2023KJ123).

**Competing interests:** The authors have declared that no competing interests exist.

# 1. Introduction

## 1.1. Research background

Prestressed concrete beams, as the core components of modern bridge engineering, have two main technical systems for prestressing application: pretensioned and posttensioned. Although posttensioned construction technology can achieve curved reinforcement distribution, it requires complex procedures such as reserving ducts and grouting anchoring, leading to inherent deficiencies such as cumbersome construction processes, high material consumption, non-compact grouting in ducts, steel corrosion, and local damage to the concrete at the anchor zone. In contrast, pretensioned construction technology transfers force through the bond between concrete and prestressed tendons, offering significant advantages such as simplified procedures, no need for anchors, and outstanding cost-effectiveness, making it particularly suitable for the production of standardized prefabricated components [1–3].

In the conventional pretensioned construction technology, the commonly used arrangement form of linear prestressed tendons does not accurately follow the actual bending moment distribution curve of structural components. Therefore, the distribution of prestress on various sections of the component does not match the corresponding external load effects optimally, especially in the beam end area where this mismatch is more pronounced, leading to the generation of oblique cracks at the beam ends. Under the condition of large span, the imbalance of stress distribution caused by linear prestressed tendons is more significant, which seriously limits the potential of span expansion of prestressed concrete beams by pretensioning method.

As a new type of prestressed tendons arrangement form of pretensioned concrete beam, the broken line pretensioned construction technology offers a potential solution to these challenges. This technology subjectifies the concrete near the beam end to pre-shear, effectively controls the generation of diagonal cracks at the beam end, enhances the section bending strength, reduces the thickness of the web plate and the amount of ordinary steel bars, and improves the mechanical properties of the member. These advantages make this technology suitable for long-span bridges [4–6].

## 1.2. Research status

At present, the research on the broken line pretensioned technology is primarily focused on theoretical calculation, experimental testing, and numerical simulation.

In terms of theoretical analysis, Lin [7] proposed an equivalent load method, which laid a theoretical foundation for the mechanical analysis of folded pre-tensioned beams. Cole [8] theoretically analyzed the prestress loss of concrete beams using the folded pretensioned method and presented a model for calculating prestress loss. Xie et al. [9] measured the compressive strains of concrete in the end area of pretensioned concrete beams, analyzed the variation of compressive strains of concrete in the transfer length scope, and verified the calculation methods of the transfer length in the codes. Huang et al. [10] established the 3D nonlinear finite element model of the pretensioned prestressed concrete beam and discussed the changing regularity of the local stress

distribution by the corner radius. Aiming at stress concentration phenomenon at prestressed bending position of the pre-tensioned concrete T-beam, Zhu et al. [11] discussed the effects of the longitudinal, vertical, transverse reinforcement ratio of the T-beam and the bending radius on the stress concentration near the bending point. Zhang et al. [12] advanced the calculation method for transverse distribution coefficients by considering the impact of diaphragm beam spacing on main beams. Wang et al. [13] proposed an innovative pulley steering gear-based formula for calculating prestressing force loss due to the angle of the steering gear and developed a novel method for calculating the vertical force. Liu et al. [14] proposed a construction scheme for prestressed tendons to significantly reduce the overturning moment and bending moment.

In terms of experimental tests and numerical simulation, Wang et al. [15] fitted the creep coefficient equation of experimental beams by long-term loading and mid-span deflection monitoring and compared it with the specification mode. Li et al. [16] conducted the fatigue behavior test and the last statical test after the fatigue behavior test, and studied the fatigue behavior of the prestressed concrete beams with pretensioned bent-up tendons. The results showed that the residual bearing capacity was not affected by the loading history and the fatigue behavior of the beam was excellent. Wang et al. [17] verified the consistency of the dynamic characteristics of I-beams with the theoretical values by means of dynamic load tests. Guo et al. [18] analyzed the effect of stress sequence on stress concentration by static load test. Liu et al. [19] pointed out that the pretensioned double-tee girder could meet the requirements of the current bridge design codes in terms of stiffness, crack resistance, and ultimate flexural strength through the full-scale tests. Liu et al. [20] investigated the bending performance of Bulb-T girder by finite element software simulation, and proposed the measures and methods to improve the overall bending performance. Wang et al. [21] presented an improved new tensioning pedestal-pile plate tensioning pedestal suitable for the prefabricated I-beam of the fold-line pretensioned method. Zhang et al. [22] found that the web thickness had less influence on the main mechanical indexes of the folded prestressed T-beam by prestressing method, but greater influence on the upper limit of the shear resistance of the section.

With the development of science and technology, artificial intelligence technology and neural network algorithm are increasingly applied to the construction and detection of bridge engineering [23]. Liu [24,25] proposed a novel physics-informed neural network approach for nonlinear structural system identification and demonstrated its application in multi-physics cases where the damping term was governed by a separated dynamics equation. Mahmoudabadi and Al-Sayegh [26,27] uses artificial neural networks to predict the bearing capacity of concrete components.

In summary, despite the significant progress made in previous studies, there are certain limitations to current achievements: Firstly, experimental subjects are predominantly rectangular and T-shaped beams, with insufficient research on I-beams that offer superior section efficiency. Secondly, the span lengths of test beams are generally less than 20 m, lacking systematic studies on the elastoplastic evolution and ultimate load-bearing capacity of large-span prestressed concrete structures, especially for the standard design span of 30 m which can effectively balance structural economy and engineering rationality while meeting the traffic needs [28,29]. Additionally, the impact of different types of broken line prestressed tendon arrangement on the mechanical performance of prestressed I-beams has been rarely explored.

In this paper, a 30 m broken line pretensioned prestressed concrete I-beam (briefly referred to as I-beam) was taken as the research object, and the on-site static load tests and numerical simulations were conducted. The mechanical properties, damage and failure process, and ultimate bearing capacity of the I-beams in the elastic-plastic stage were studied. On this basis, the optimized arrangement of prestressed tendons was discussed. The research results can provide reference for the design and construction of the same types of I-beams.

## 2. On-site static load test

### 2.1. Introduction of I-beam

As shown in Fig 1, the span of the I-beam is 30 m and the height is 1.80 m. Partial prestressed tendons are arranged at the bottom of the beam, while partial prestressed tendons(N1-N5) are bent at 3 m from the center line of the beam. The

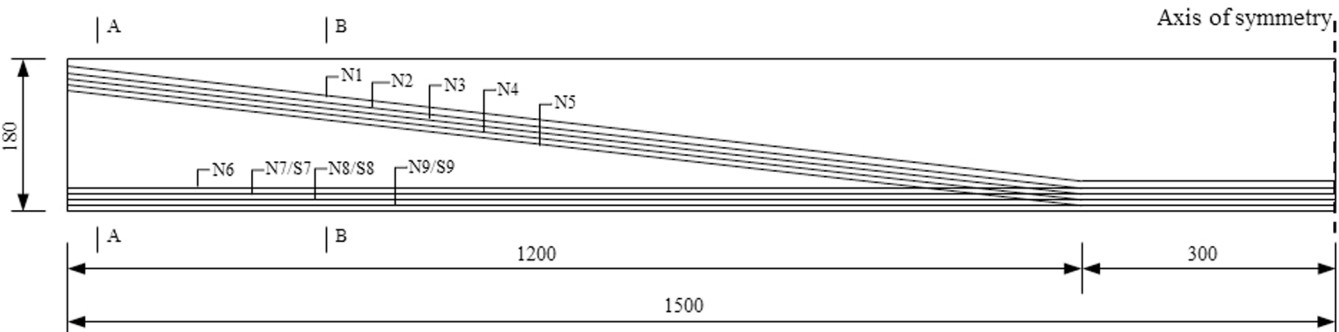

**Fig 1. Layout diagram of prestressed tendons (unit: cm).**

cross-sectional dimensions of the I-beam are shown in Fig 2. The widths of the upper and lower flanges are 1.40 m and 0.95 m respectively. The thickness of the web plate at the end section is 0.3 m, and that at the mid span section is 0.2 m. 36 prestressed tendons are arranged in the I-beam, with a nominal diameter of 15.2 mm. The strength grade of concrete is C60, and tensile strength of the prestressed tendons is 1860 MPa.

## 2.2. Setup of test sensor

Under the action of self-weight load and normal operation and maintenance process, the mid-span section of I-beam will bear significant internal forces. Referring to the relevant specification [30], In the mid-span and 1/4-span sections, ten strain sensors (S 1-1 to S 1-5, S 2-1 to S 2-5) were arranged to measure the longitudinal linear strain. In addition, two displacement sensors (D-1, D-2) were arranged at the bottom of the mid-span section to measure the vertical displacement during the loading process. The mechanical properties of I-beam are analyzed by test data, and the specific arrangement of the sensor is shown in Figs 3 and 4.

## 2.3. Test loading scheme

The static load tests of the I-beam are carried out under the approximate simply supported restraint condition, with the loading point situated at the mid-span cross-section of the I-beam. Due to the limited conditions of the construction site, it

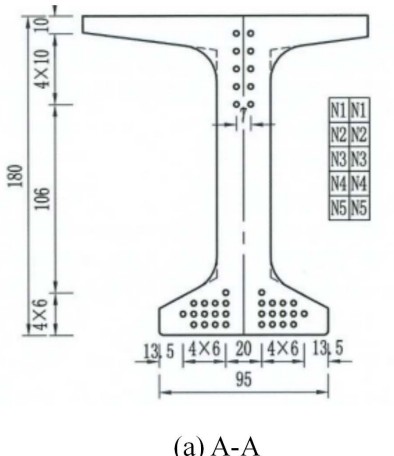

(a) A-A

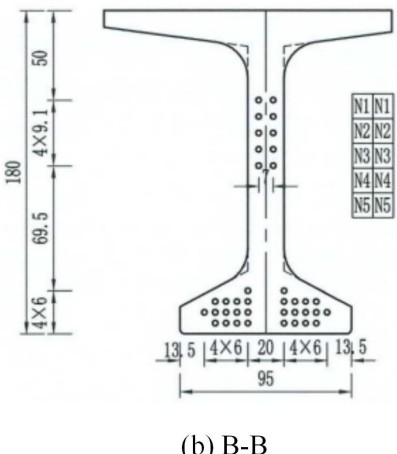

(b) B-B

**Fig 2. Cross section of I-beam (unit: cm).**

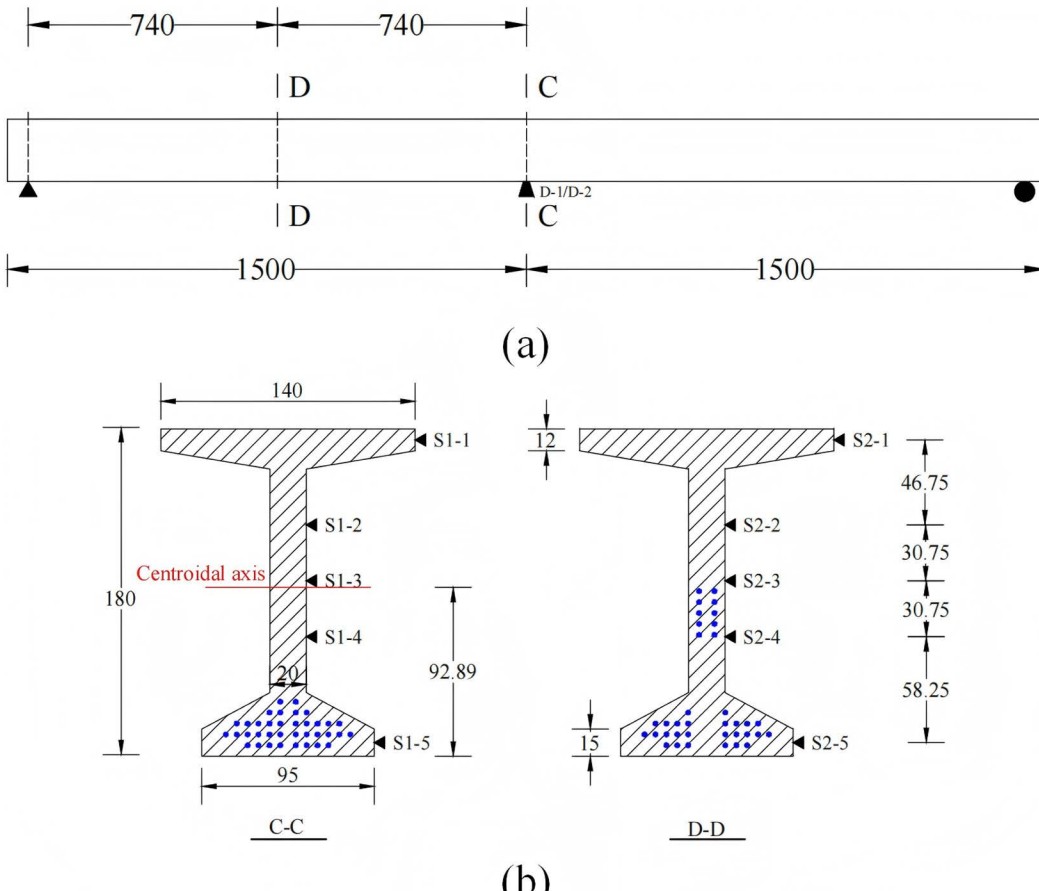

**Fig 3. Schematic diagram of sensor arrangement (unit: cm): (a) Arrangement of displacement sensors; (b) Arrangement of strain sensors.**

is difficult to use traditional testing methods for loading. Therefore, an I-beam with the same specification as the test beam is used as the loading beam, and a hydraulic jack is arranged between the end of the loading beam and the test beam to achieve concentrated force loading (shown in Fig 5). The on-site loading situation is shown in Fig 6 and the specific loading method is outlined as follows:

(1) The steel plate and hydraulic jack are arranged at the top of the mid-span of the test beam, and the loading beam is hoisted. One end of the loading beam is moved to the middle of the test beam.

(2) The end of the loading beam slowly drops to 1~2 cm away from the hydraulic jack.

(3) The oil pump is started to make the loading beam contact with the roof of the hydraulic jack.

(4) As shown in Table 1, the concentrated load is gradually applied. The load is held for 5 minutes at each condition, and unloaded after the completion of the loading.

Due to the space limitation of this paper, the static load test data are described and analyzed in the following comparison with the numerical simulation results, and will not be repeated here.

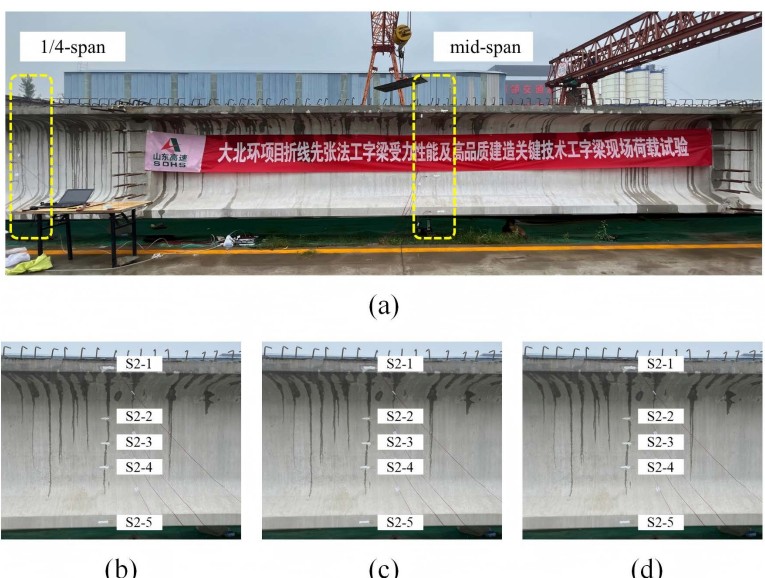

Fig 4. Arrangement of sensors: (a) Overall view of I-beam; (b) Strain sensors in 1/4-span; (c) Strain sensors in mid-span; (d) Displacement sensors.

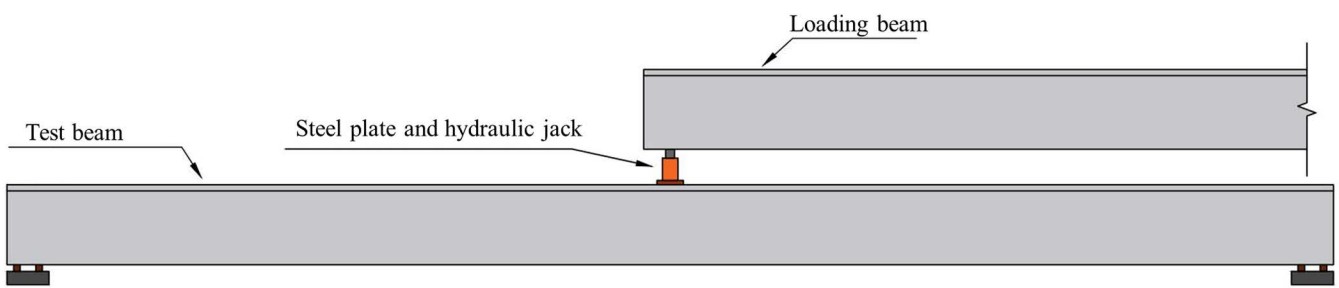

Fig 5. Loading diagram.

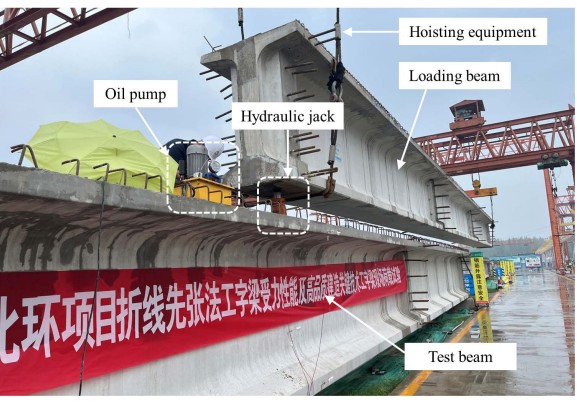

Fig 6. Loading site of I-beam.

**Table 1. Loading conditions.**

| Condition number | Pressure of hydraulic jack/MPa | Corresponding static load/t |
|---|---|---|
| 1 | 5 | 8.0 |
| 2 | 10 | 15.5 |
| 3 | 12 | 19.0 |
| 4 | 14 | 22.0 |
| 5 | 16 | 25.0 |
| 6 | 18 | 28.0 |
| 7 | 19 | 29.5 |
| 8 | Unload | |

## 3. Numerical models

In this paper, the ABAQUS software is utilized to establish a finite element model of the test beam and to perform a numerical simulation of the static load test. The material parameters of the model are determined according to the material performance test, and the basic parameters are shown in Tables 2 and 3. In the element selection of the model, the C3D8R element is used to simulate the concrete, B31 element is used to simulate the prestressed tendons, and the T3D2 element is used to simulate the ordinary steel bars. In the selection of constitutive model of materials, the concrete is simulated by concrete damaged plasticity model, and the prestressed tendons and ordinary steel bars are simulated by ideal elastoplastic model. Considering the calculation accuracy and efficiency, the element size of concrete, prestressed tendons and ordinary steel bars of I-beam is determined to be about 10 cm. Since the loading pad in the mid-span is not the main research focus, the element size is determined to be 50 cm.

To study the bond-slip behavior between concrete and prestressed tendons, a bond-slip interface layer is set between concrete and prestressed tendons. The interface layer is discretized by three-dimensional eight-node cohesive element (COH3D8), and the user subroutine developed by the Fang [31] is used to realize the nonlinear bond-slip constitutive relationship under external load. The ordinary steel bars are embedded into the concrete to achieve collaborative deformation, which does not consider the bond-slip behavior between ordinary steel bars and concrete. The numerical analysis model is shown in Fig 7, and the ideal simple support constraint is applied to the I-beam model. One end of the beam constrains the linear displacement in three directions, and the other end constrains the linear displacement in two directions except the axis direction.

In the finite element numerical calculation, there are two main equivalent simulation methods for prestressed load: the equivalent load method and the decreasing temperature method [32]. Compared with the equivalent load method, the decreasing temperature method is a commonly used method to apply prestress in ABAQUS. By reducing the temperature

**Table 2. Relevant parameters of concrete.**

| Material type | Elastic modulus/MPa | Density /(kg·m⁻³) | Poisson's ratio | Dilation Angle/° | Eccentricity | $f_{b0}/f_{c0}$ | K | Viscosity parameter |
|---|---|---|---|---|---|---|---|---|
| Concrete (60) | $3.6 \times 10^4$ | 2400 | 0.2 | 30 | 0.1 | 1.16 | 0.667 | 0.005 |

**Table 3. Relevant parameters of prestressed tendons and steel bars.**

| Material type | Elastic modulus/MPa | Density/(kg·m⁻³) | Poisson's ratio |
|---|---|---|---|
| Prestressed tendons | $1.95 \times 10^5$ | 7850 | 0.3 |
| Ordinary steel bars (HRB400) | $2.0 \times 10^5$ | 7850 | 0.3 |

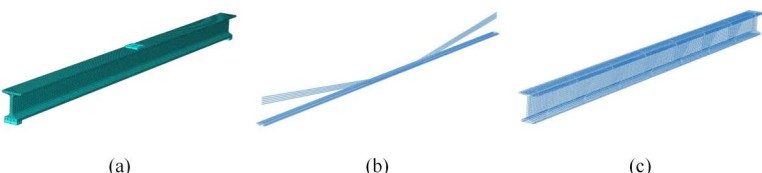

(a)                    (b)                    (c)

**Fig 7. Numerical analysis model: (a) Concrete; (b) Prestressed tendons; (c) Ordinary steel bars.**

of the prestressed tendon unit, the shrinkage force generated by the shrinkage of the prestressed tendon is used to simulate the effect of the prestressed tendon on the concrete. The magnitude of the prestress can be controlled by the cooling value, which is calculated by the principle that the linear strain generated by the temperature is equal to the linear strain generated by the axial force. The decreasing temperature method can simulate the prestress in complex concrete structures with high accuracy and ease of use. Therefore, this paper simulates the prestress through the decreasing temperature method, and the calculation formula is as follows [33,34].

$$T = \frac{F_T}{\alpha \times E_s \times A_s}$$

(1)

where $T$ is the temperature value, $F_T$ is the control tension of prestressed tendons, $E_s$ is the elastic modulus, $A_s$ is the cross-sectional area, and $\alpha$ is the linear expansion coefficient, which is set to $1.2 \times 10^{-5}$.

## 4. Comparative analysis of numerical simulation and model test data

### 4.1. Comparison of arching height

The application of prestress results in the occurrence of pre-arching of the I-beam, and the arching height of the mid-span cross-section of the I-beam has been measured to be 22.90 mm following on-site machining. In the numerical simulation, the arching height of the mid-span cross-section is 24.13 mm (shown in Fig 8), and the relative error is 5.37% compared to the test data. The numerical analysis model has a good agreement with the on-site test model.

### 4.2. Comparison of strain data

The static load tests were simulated by finite element analysis, and the strain data of each measuring point were compared with the test data. The comparisons of "load-strain" curves are shown in Figs 9 and 10.

According to the analysis of test data, with the increase of concentrated load in the mid-span section, the strain value at each measurement point exhibits an overall linear growth trend. The measuring points 1–3, arranged in the middle and upper

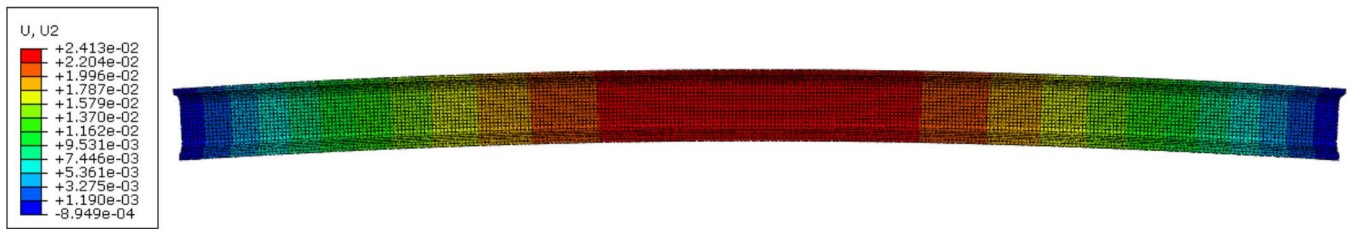

**Fig 8. Arching height of numerical analysis model.**

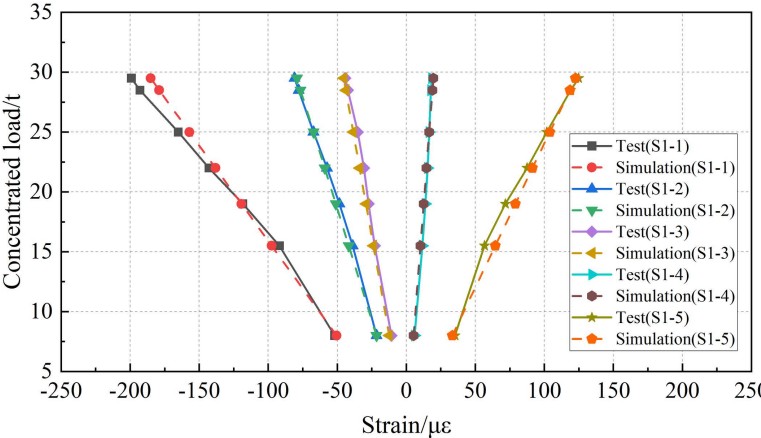

**Fig 9. Comparison of strain data at mid-span section.**

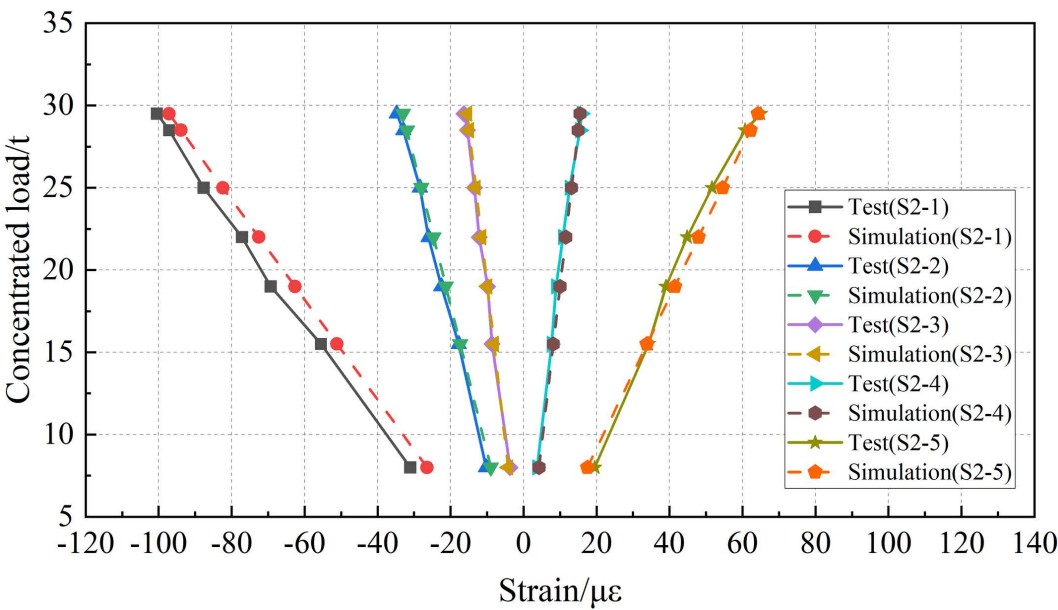

**Fig 10. Comparison of strain data at 1/4-span section.**

regions of the mid-span and 1/4-span sections are under compression, while the measuring points 4–5, arranged in the lower part are under tension. Under the action of condition 7, the maximum tensile strain at the mid span section was 124.8 $\mu\varepsilon$, and the maximum compressive strain was −199.1 $\mu\varepsilon$. At the 1/4-span section, the maximum tensile strain was 64.9 $\mu\varepsilon$, and the maximum compressive strain was −100.5 $\mu\varepsilon$. During the on-site static load tests, the I-beam remained in the elastic stage.

According to the comparison of the two analysis methods, the numerical simulation curve has a good agreement with the test curve, and there are some differences between the simulation results and test data under individual conditions. To further quantify the differences between the two analysis methods and verify the rationality of the numerical simulation method, the relative error between the numerical simulation and the test data is calculated. The statistical results are shown in Fig 11.

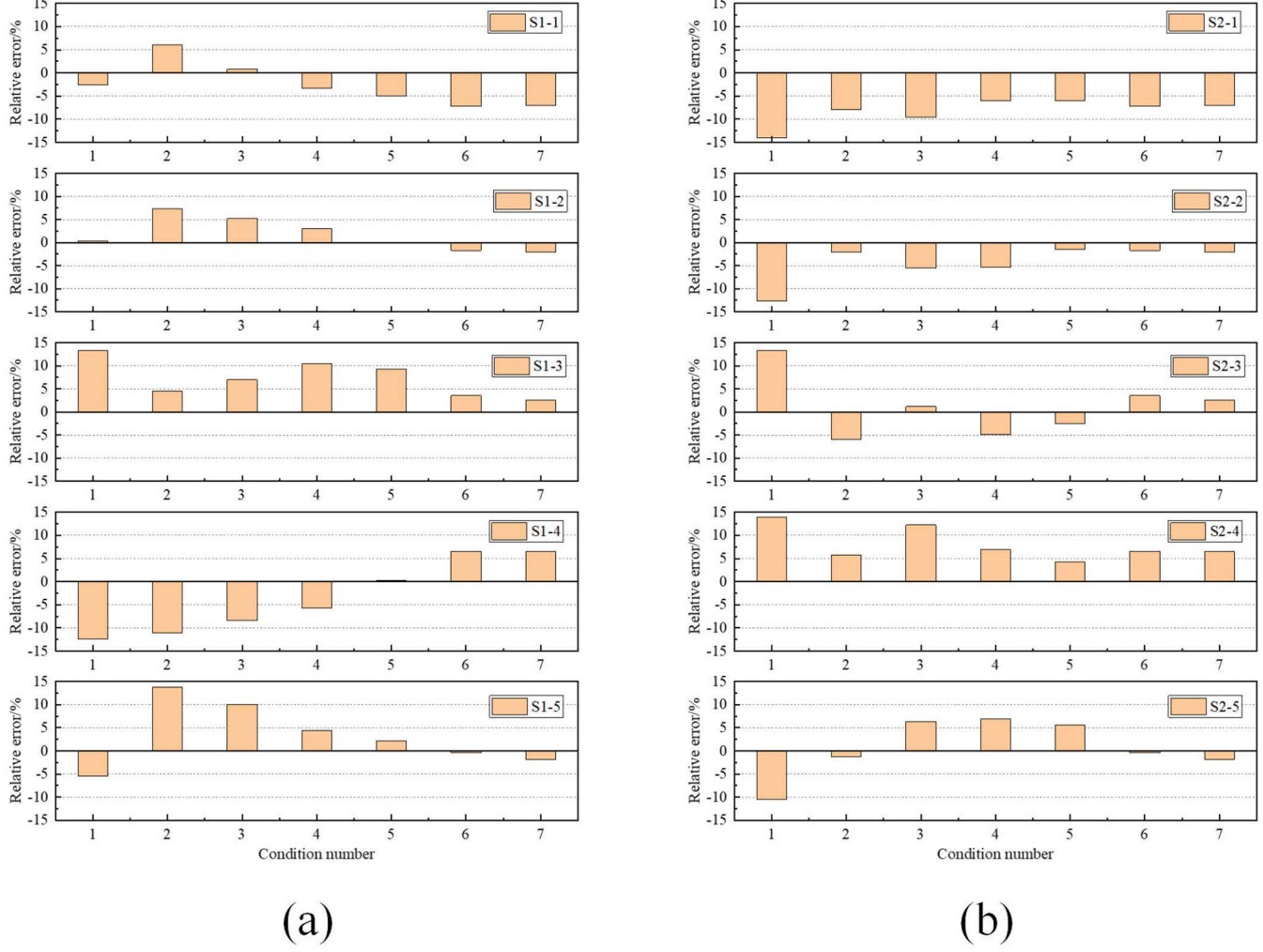

## (a)                    (b)

**Fig 11. The relative error between the numerical simulation and the test data: (a) Mid-span section; (b) 1/4-span section.**

It can be seen that when the concentrated load is smaller, the relative error is relatively large, especially under the action of condition 1. As the concentrated load increases, the relative error shows a decreasing trend. During the loading process, the maximum relative error at the mid-span section is 13.72%, and that at the 1/4-span section is 14.09%. The main reasons for the error are as follows:

(1) The materials in the numerical analysis model adopt idealized assumptions, which do not fully match the actual materials.

(2) The damage plasticity model parameters of concrete in the numerical analysis model are determined according to the "Code for Design of Concrete Structures" [35], which are not completely consistent with the actual materials.

(3) The numerical analysis model is an ideal simply supported beam. However, the static load tests are limited by the on-site conditions, and the two ends of the I-beam are placed on rigid blocks, which cannot achieve the ideal simply supported condition.

(4) The loading device and accuracy of at the test site cannot be completely consistent with the numerical simulation.

Overall, the relative error of most measuring points is controlled within 10%. In the simulation of strain response, the numerical analysis method is reasonable and reliable.

### 4.3. Comparison of displacement data

The vertical displacement of the mid-span section obtained by numerical simulation is compared with the test data, and the relative errors are calculated. The statistical results are shown in Table 4.

It can be seen that with the increase of the concentrated load, the vertical displacement of the mid-span section basically shows a linear growth trend. Under the action of condition 1, the relative error between numerical simulation and test is the largest, which is 10.59%. In addition, the relative errors under other conditions are less than 5%. Under the action of condition 7, the maximum vertical displacement at the mid-span section measured by static load test is 12.21 mm, and that obtained from numerical simulation is 11.89 mm. In the simulation of displacement response, the numerical analysis method is reasonable and reliable.

### 4.4. Comparison of neutral axis position

The presence of prestressed tendons can cause the change of the neutral axis position in the cross-section. To determine the neutral axis position, the strain data at different heights of the mid-span section were extracted, and the strain distribution of the I-beam section was analyzed. Taking conditions 1, 3, 5 and 7 as examples, the strain data of each measuring point at the mid-span section and the fitted "height-strain" curve are shown in Fig 12.

It can be seen that within the elastic range, the strains of the five measuring points at the cross section are basically linearly distributed along the height. The neutral axis position can be obtained through the fitted formulas in Fig 12, and the height of the neutral axis from the bottom of the cross section is listed in Table 5.

Under various loading conditions, the straight lines fitted based on the test data and the simulation results basically coincide, and the relative errors of the neutral axis height are all less than 5%. Fig 3 demonstrates that the height of centroid axis from the bottom is 92.89 cm. The presence of prestressed tendons causes pre-pressure on the I-beam section, and the neutral axis moves downwards. In the design of the structure, the pre-pressure can be controlled to offset part of the tensile stress in the tension zone of the I-beam and improve the bending performance of the beam body.

## 5. Ultimate flexural capacity and optimized arrangement of prestressed tendons

Due to limitations in on-site test conditions, it is impossible to continue loading and observing the failure mode of the I-beam. Based on the above finite element analysis method verified by test data, further theoretical analysis and numerical simulation were carried out to study the ultimate flexural capacity of the structure. On this basis, the optimized arrangement of prestressed tendons is analyzed.

**Table 4. Relative error of displacement response.**

| Condition number | Test data/mm | Simulation result/mm | Relative error/% |
| --- | --- | --- | --- |
| 1 | 3.59 | 3.21 | −10.59 |
| 2 | 5.93 | 6.22 | 4.89 |
| 3 | 7.37 | 7.62 | 3.39 |
| 4 | 8.76 | 8.84 | 0.91 |
| 5 | 10.22 | 10.07 | −1.47 |
| 6 | 11.64 | 11.51 | −1.12 |
| 7 | 12.21 | 11.89 | −2.62 |

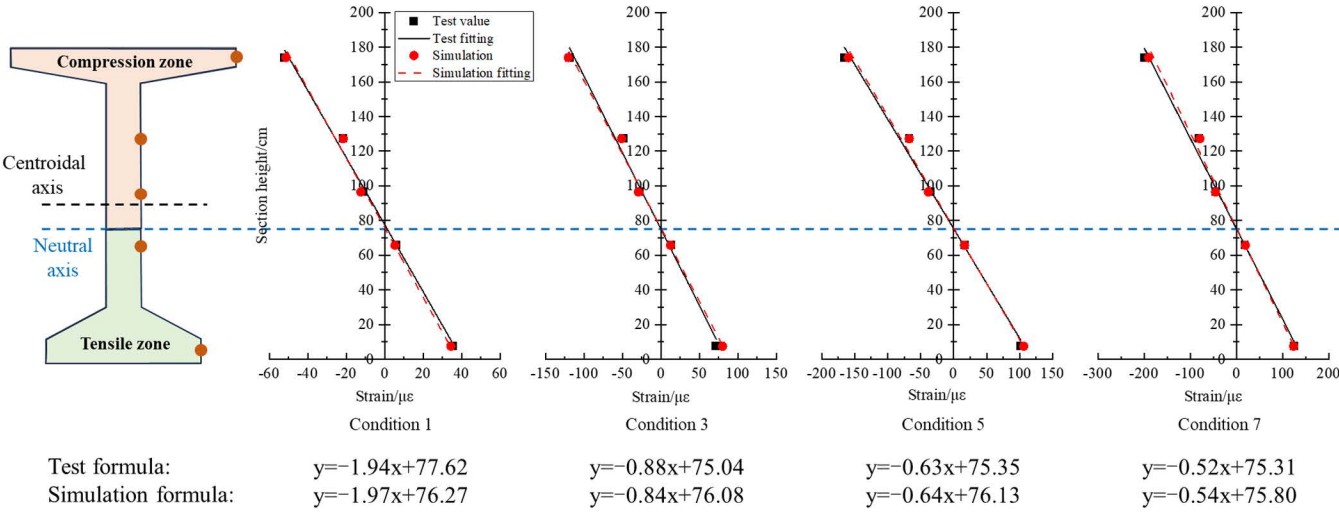

| | Test formula: | Simulation formula: |
|---|---|---|
| | y=−1.94x+77.62 | y=−1.97x+76.27 |
| | y=−0.88x+75.04 | y=−0.84x+76.08 |
| | y=−0.63x+75.35 | y=−0.64x+76.13 |
| | y=−0.52x+75.31 | y=−0.54x+75.80 |

**Fig 12. Strain distribution curve at the mid-span section.**

**Table 5. Neutral axis position at the mid-span section.**

| Condition number | 1 | 3 | 5 | 7 | Average value |
|---|---|---|---|---|---|
| Neutral axis height from the bottom (test)/cm | 77.62 | 75.04 | 75.35 | 75.31 | 75.83 |
| Neutral axis height from the bottom (simulation)/cm | 76.27 | 76.08 | 76.13 | 75.80 | 76.07 |
| Relative error/% | −1.74 | 1.39 | 1.04 | 0.65 | 0.32 |

### 5.1. Ultimate flexural capacity

**5.1.1. Theoretical analysis.** Referring to the relevant specification [35], the cross section of the I-beam is shown in Fig 13. Without considering the influence of ordinary concrete and ordinary steel bars in the tension zone on the flexural bearing capacity, the flexural bearing capacity of the section can be calculated according to the formula (2):

$$M = \alpha_1 f_c (b'_f - b) h'_f (h_p - \frac{h'_f}{2}) + \alpha_1 f_c b x (h_p - \frac{x}{2})$$

(2)

Where, $\alpha_1$ is calculation coefficient (For C60 concrete, $\alpha_1$ is 0.98), $x$ is the compression zone height, $b'_f$ is the width of the flange in the compression zone of the I-beam, $b$ is the width of the web, $h'_f$ is the height of the flange in the compression zone of the I-beam, $h_0$ is the effective height of cross-section, $A_p$ is the cross section area of the prestressed tendons in the tension zone, $A_s$ is the cross section area of the ordinary steel bars in the tension zone, $f_c$ is the design value of axial compressive strength of concrete, $f_y$ is the design value of tensile strength of ordinary steel bar, $f_{py}$ is the design value of tensile strength of prestressed tendon.

According to the height of neutral axis obtained by fitting the test data, the ultimate flexural capacities is 13649.85 kN·m.

**5.1.2. Numerical simulation.** For the prestressed reinforced concrete beams, it is considered that when one of the following states is reached, the prestressed concrete I-beam is considered to reach the ultimate state of bearing capacity [36]:

[1] The maximum compressive strain of concrete reaches $\varepsilon_{cu}$;

[2] The prestressed tendon reaches yield state;

[3] The deflection of the mid-span section exceeds the limit value.

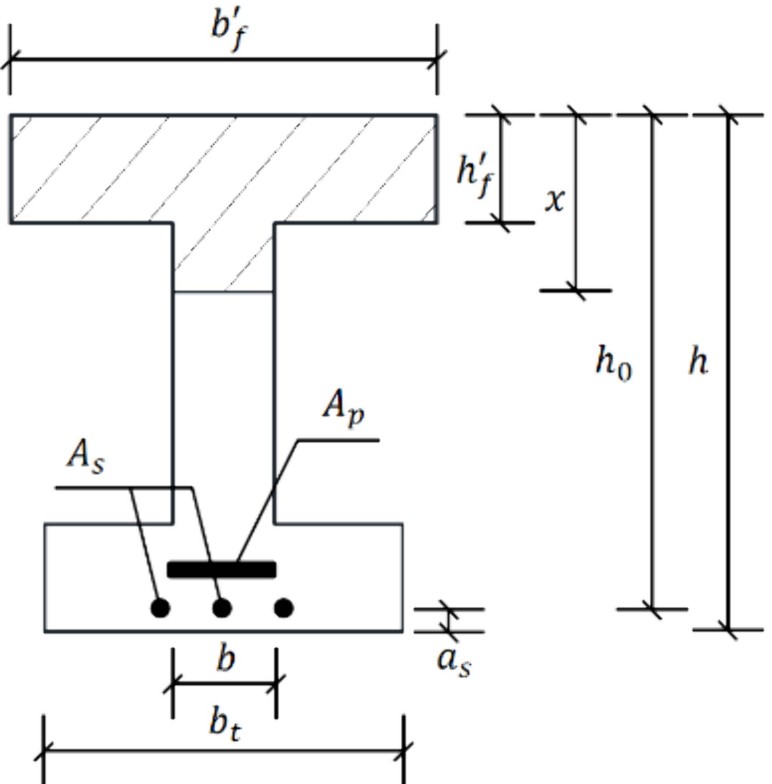

**Fig 13. Schematic diagram of cross section of the I-beam.**

According to the relevant code [37], the ultimate compressive strain of concrete $\varepsilon_{cu}$ can be calculated by the following formula:

$$\varepsilon_{cu} = 0.0033 - (f_{cu,k} - 50) \times 10^{-5} \tag{3}$$

Where, $f_{cu,k}$ is the standard value of compressive strength of concrete cube. The ultimate compressive strain $\varepsilon_{cu}$ of C60 concrete is calculated to be 0.0032.

In the numerical simulation, the gradually increasing concentrated loads were applied in the mid-span section of the finite element model to study the mechanical properties and failure mode of the I-beam. The "load-displacement" curve at the mid-span section is shown in Fig 14.

It can be seen that the whole loading process is divided into three stages:

(1) Elastic stage: In the initial stage of loading, the "load-displacement" curve is basically a straight line, and the I-beam remains intact with small deformation.

(2) Yield stage: When the concentrated load reaches 1290 kN, the local concrete at the bottom of the I-beam cracks under tension. The "load-displacement" curve has a turning point, and the tangent slope of the curve gradually decreases. When the concentrated load reaches 1832 kN, some ordinary steel bars in the tensile zone yield, and the displacement at the mid-span section further increases.

(3) Failure stage: When the concentrated load reaches 1910 kN, the individual prestressed tendons at the bottom of the I-beam enter the yield state. At this time, the maximum compressive strain of the concrete is 0.003, which is close to the ultimate value.

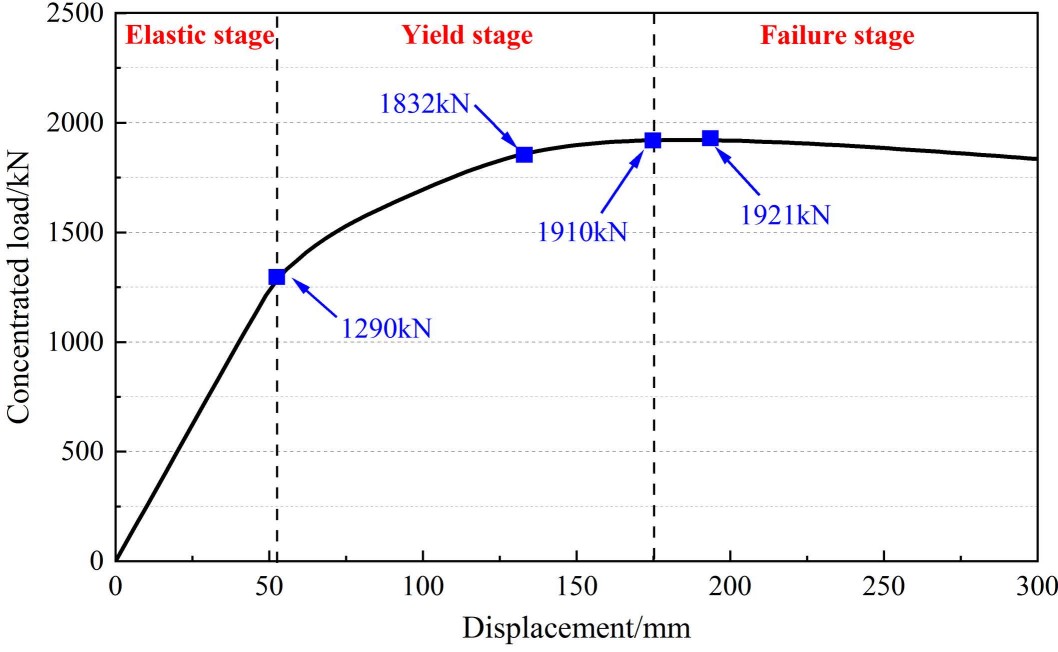

**Fig 14. "Load-displacement" curve at the mid-span section.**

The vertical displacement of the mid-span section is 175 mm, which does not exceed the specification limit. After further loading, the prestressed tendons yield successively. The vertical displacement of the mid-span section increases significantly, while the load remains basically unchanged. When the concentrated load reaches 1921 kN, the I-beam cannot continue to bear the load.

Therefore, it is considered that the prestressed tendons play a controlling role in the ultimate bearing capacity of the I-beam. In the process of numerical simulation, the damage situation of the I-beam at each stage is shown in Fig 15.

When the prestressed tendons yield, the ultimate bending moment of the mid-span section is 14134.00 kN·m, and it is 3.55% larger than the theoretical analysis result. The reason is that the theoretical calculation formula does not consider the contribution of ordinary concrete and ordinary steel bars in the tension zone to the bearing capacity, and the nonlinear behavior of the material, which is conservative.

## 5.2. Arrangement of prestressed tendons

In practical engineering, the common arrangement scheme of prestressed tendons is the two-fold point method. As shown in Fig 16, two benders are set on both sides of the I-beam, and the prestressed tendons are bent at the bender. The advantage of this arrangement scheme is that the number of benders is small, the construction is simple and economical, but the stress concentration is obvious.

Considering the distribution of bending moment and shear force of I-beam, a four-fold point method was designed. As shown in Fig 16, the prestressed tendons numbered N1~N3 were bent at 3 m from the center line of the beam, and the prestressed tendons numbered N4~N5 were bent at 6 m from the center line of the beam. Focusing on the arching height and ultimate flexural capacity of I-beams, numerical simulations were conducted.

Using the four-fold point method to arrange the prestressed tendons, the arching height of the I-beam after tension is 26.49 mm, which is 9.78% higher than that using the two-fold point method. Although the arching height increases, it still meets the design requirements. In addition, a larger arching height may bring better bearing capacity.

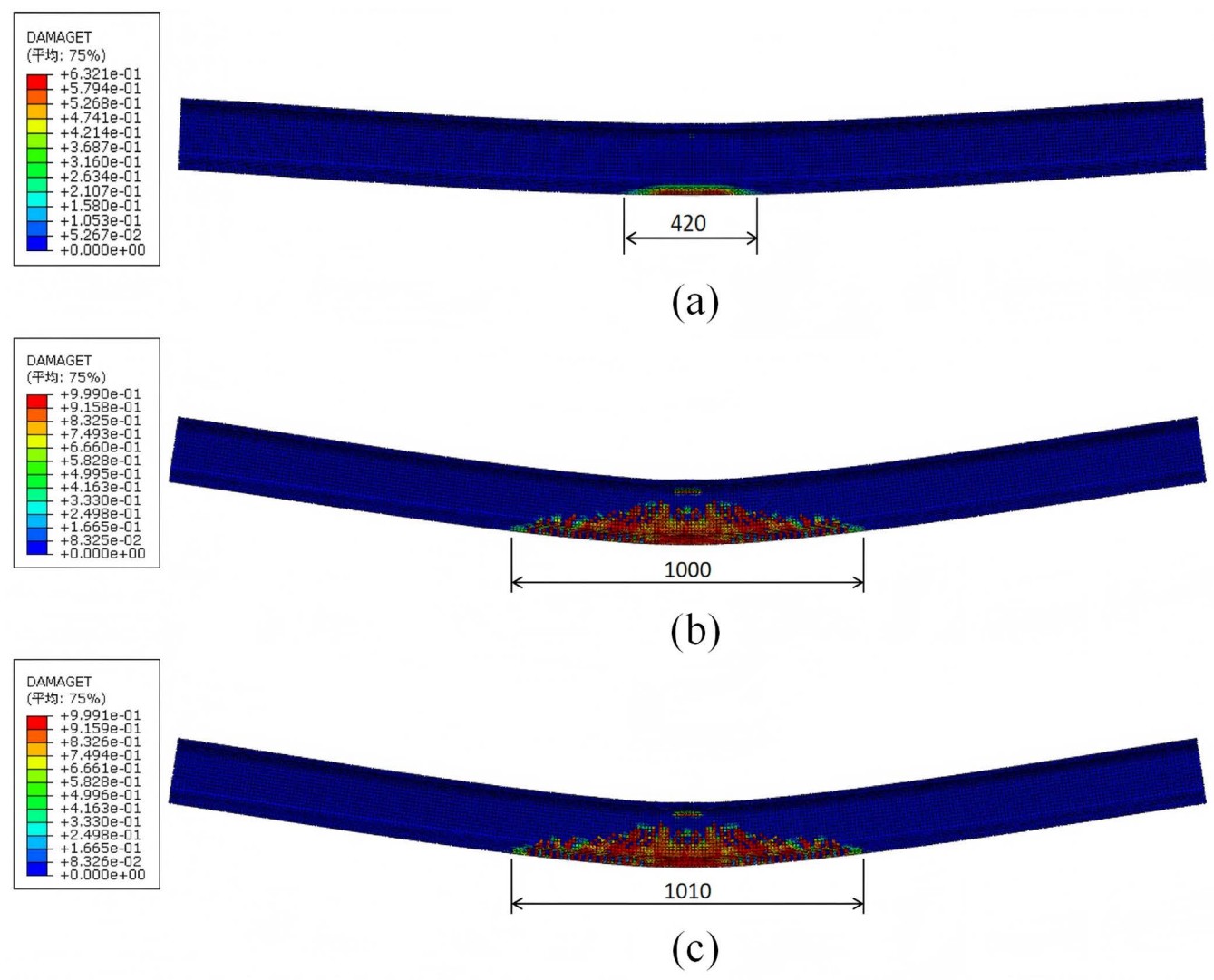

**Fig 15. Damage situation of I-beams at different stages (unit: cm): (a) Local concrete at the bottom cracks; (b) Individual prestressed tendons enter the yield state; (c) Peak load state.**

The gradually increasing concentrated loads were applied in the mid-span section of the I-beam with prestressed tendons arranged using the four-fold method to study the damage of the structure. The "load-displacement" curve was extracted and compared with the I-beam with the two-fold method, as shown in Fig 17.

It can be seen that the mechanical properties of I-beams with different arrangements of prestressed tendons are basically the same in the elastic stage. Using the four-fold point method to arrange the prestressed tendons, the cracking load and the corresponding displacement of the concrete at the bottom of the I-beam are basically consistent with the two-fold point method.

After entering the yield stage, the "load-displacement" curves of the two methods deviate, and the bearing capacity of I-beams using the four-fold point method is higher than that of the two-fold point method. The curve slope of four-fold point method $K_4$ is slightly larger than that of two-fold point method $K_2$. When the vertical displacement is 75 mm, $K_4$ is 9.03 kN/mm,

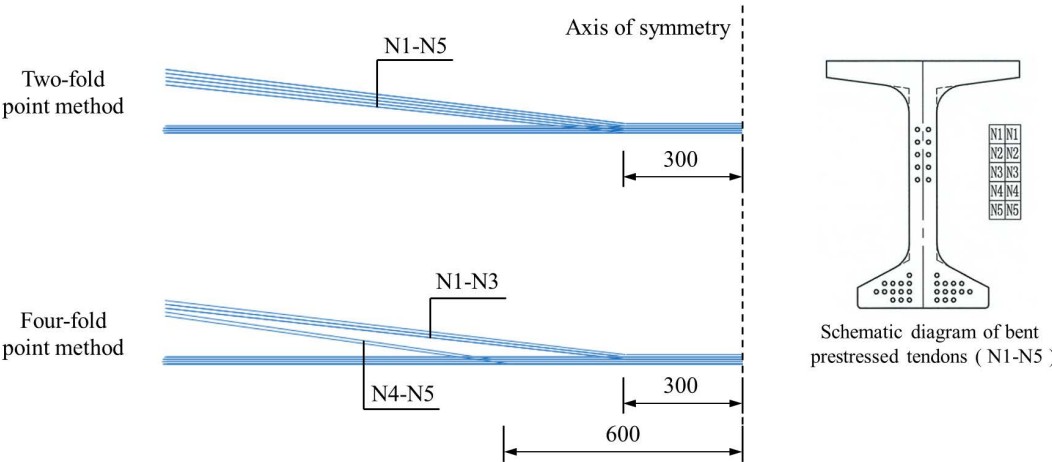

**Fig 16. Arrangement of prestressed tendons (unit: cm).**

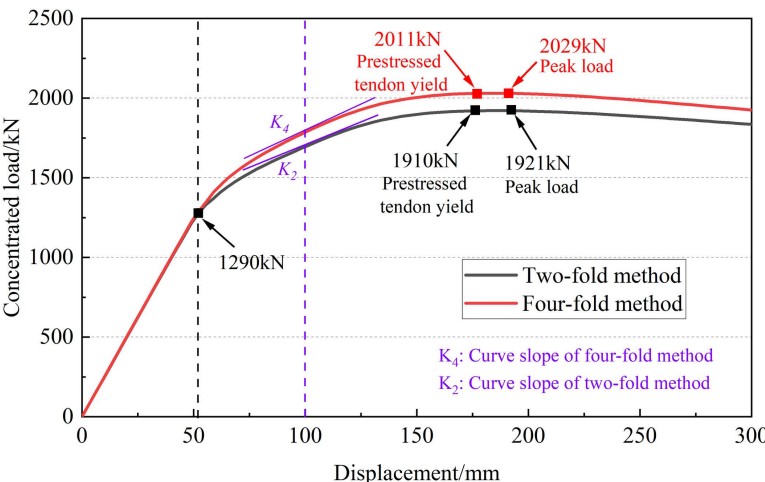

**Fig 17. Comparison of "load-displacement" curves.**

which is 14.60% higher than $K_2$ of 7.88 kN/mm. When the vertical displacement is 100 mm, $K_4$ is 6.57 kN/mm, which is 8.96% higher than $K_2$ of 6.03 kN/mm. When the prestressed tendon yields, the concentrated load on the I-beam of the four-fold point method is 2011 kN, which is 5.3% higher than that of the two-fold method. In addition, the peak load of the four-fold point method is 2029 kN, with an increase of 5.6%. The main reason is that using the four-fold point method to arrange the prestressed tendons can make the geometry of the prestressed tendons closer to the actual bending moment envelope diagram of the I-beam. The four-fold point method can provide higher stiffness and bearing capacity to the I-beam after concrete cracking.

## 6. Conclusions

This paper takes a 30 m broken line pretensioned prestressed concrete I-beam as a research object. On-site static load test, numerical simulation and theoretical analysis are conducted to study the mechanical performance, failure mode, ultimate flexural capacity and optimized arrangement of prestressed tendons. The conclusions are as follows:

(1) In the on-site static load tests, the strain values at the mid-span section and 1/4-span basically show a linear growth trend, and the I-beam is always in the elastic stage. The numerical simulation results are in general agreement with the test data with regard to strain, displacement, and neutral axis position, with minor discrepancy.

(2) When the concentrated load reaches 1910 kN, the prestressed tendons at the bottom of the I-beam enter the yield state, and the prestressed tendons play a controlling role in the ultimate bearing capacity of the I-beam. The numerical simulation result of the ultimate bending moment in the mid-span section is 3.55% larger than the theoretical analysis result, and the theoretical calculation formulas tend to be conservative.

(3) When the prestressed tendons are arranged by the four-fold point method, the I-beam has higher stiffness and bearing capacity after concrete cracking.

The above conclusions can serve as a reference for the design and construction of broken line pretensioned prestressed concrete I-beam. Further research can be carried out on the revision of the theoretical formula for the ultimate flexural capacity and the optimal design of the cross-section. Physics informed neural network and artificial intelligence technology can be also used to carry out intelligent detection of bridge structures and propose an optimized design scheme for the I-beam.

## Supporting information

**S1 File. Mechanical behavior of concrete.**
(ZIP)

## Acknowledgments

This research was funded by the Shandong Provincial Natural Science Foundation and the National Natural Science Foundation. The numerical calculations have been done on the supercomputer system in Shandong Jianzhu University, and the authors appreciate this help.

## Author contributions

**Conceptualization:** Peisen Wang.

**Data curation:** Chenning Song.

**Formal analysis:** Jiacheng Shi.

**Funding acquisition:** Peisen Wang, Chenning Song.

**Investigation:** Peisen Wang, Chenning Song.

**Methodology:** Jiacheng Shi.

**Resources:** Peisen Wang, Chenning Song.

**Software:** Jiacheng Shi.

**Supervision:** Jiacheng Shi.

**Validation:** Peisen Wang, Chenning Song.

**Visualization:** Jiacheng Shi.

**Writing – original draft:** Jiacheng Shi.

**Writing – review & editing:** Jiacheng Shi.

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
