## [Decision Letter · Decision Letter 0]

Dear Dr. Song,

Thank you for submitting your manuscript to PLOS ONE. After careful consideration, we feel that it has merit but does not fully meet PLOS ONE’s publication criteria as it currently stands. Therefore, we invite you to submit a revised version of the manuscript that addresses the points raised during the review process. I invite you to resubmit your manuscript after addressing the comments given by the reviewers. 

When revising your manuscript consider all issues mentioned by the reviewers and please outline every change made in response to their comments. Please note that your revised submission may need to be re-reviewed.

We look forward to receiving your revised manuscript.

Kind regards,

Jeyashree T M, Ph. D

Academic Editor

PLOS ONE

“the Shandong Provincial Natural Science Foundation, China ( ZR2021ME238, ZR2024ME056) the National Natural Science Foundation, (51908338) the Shandong Provincial Youth Innovation Team Plan, (2023KJ123)”

“This research was funded by the Shandong Provincial Natural Science Foundation, China, grant numbers ZR2021ME238 and ZR2024ME056; the National Natural Science Foundation, grant number 51908338; and the Shandong Provincial Youth Innovation Team Plan, grant number 2023K123. The numerical calculations have been done on the supercomputer system in Shandong Jianzhu University, and the authors appreciate this help.”

“the Shandong Provincial Natural Science Foundation, China ( ZR2021ME238, ZR2024ME056) the National Natural Science Foundation, (51908338) the Shandong Provincial Youth Innovation Team Plan, (2023KJ123)”

6. We note that your Data Availability Statement is currently as follows: [All relevant data are within the manuscript and its Supporting Information files.]

7. PLOS requires an ORCID iD for the corresponding author in Editorial Manager on papers submitted after December 6th, 2016. Please ensure that you have an ORCID iD and that it is validated in Editorial Manager. To do this, go to ‘Update my Information’ (in the upper left-hand corner of the main menu), and click on the Fetch/Validate link next to the ORCID field. This will take you to the ORCID site and allow you to create a new iD or authenticate a pre-existing iD in Editorial Manager.

Reviewers' comments:

Reviewer's Responses to Questions

**Comments to the Author**

1. Is the manuscript technically sound, and do the data support the conclusions?

Reviewer #1: Yes

Reviewer #2: Yes

2. Has the statistical analysis been performed appropriately and rigorously?

Reviewer #1: Yes

Reviewer #2: Yes

3. Have the authors made all data underlying the findings in their manuscript fully available?

Reviewer #1: Yes

Reviewer #2: Yes

4. Is the manuscript presented in an intelligible fashion and written in standard English?

Reviewer #1: Yes

Reviewer #2: Yes

Reviewer #1: This paper investigates the mechanical performance and flexural capacity of a prestressed concrete I-beam. My major comments are as follows.

1. The model assumes perfect bond with no slip, which I think is inaccurate for draped or bent tendons in your case.

2. The simulation doesn't capturing the final failure modes, which might also be interesting to know.

3. More details should be included for the FEM model setup. For instance the element size, key assumption, etc.

4. it would be valuable for the authors to compare or discuss the potential of physics informed neural network (PINN)-based methods as an alternative or complementary tool to FEM-based modeling

5. Following on previous comments, I suggest to include the recent work of PINN for structural analysis. For instance "Physics-informed neural networks for system identification of structural systems with a multiphysics damping model" and "Physics-informed neural network for nonlinear structural system identification".

6. I suggest to separate the first introduction into two sections (introduction and related work)

Reviewer #2: To improve and revise the paper titled "Static load test and bearing capacity analysis of broken line pretensioned prestressed concrete I-beam" here are some constructive suggestions:

Clearly articulate the novelty of this study in the introduction. While the paper highlights that few studies exist on large-span broken line pretensioned I-beams, it should more explicitly state what distinguishes this research from previous work.

Emphasize why the 30 m span and the chosen broken line tendon arrangement are practically significant.

Reorganize the introduction to follow a clear logical flow: (i) background and existing challenges in prestressed beam design, (ii) shortcomings of linear tendon configurations, (iii) advantages of broken line configurations, (iv) the novelty and scope of the current study.

Provide a clearer description of boundary conditions and simplifications used in the FE model (e.g., support type, contact assumptions).

Elaborate on the validation process used to confirm the model accuracy, especially regarding the cooling method for prestress simulation.

Specify mesh size, element quality, and convergence criteria used in ABAQUS for numerical simulations.

The comparison between the two-fold and four-fold tendon arrangement methods could be enhanced by quantitative stiffness comparisons (e.g., change in slope of load-displacement curves).

While some references are included, the literature review could be expanded to examine recent studies, such as:

o Rapid Analysis of CFRP-Reinforced Concrete Structures Using Artificial Neural Networks.

o Prediction of load-bearing capacity of RC Columns (CWA) using Artificial Neural Networks (ANN) trained on a Hybrid Experimental Database HEXP.

o Least squares collocation method in Moho depth determination in Iran using gravity gradient data.

**Do you want your identity to be public for this peer review?** For information about this choice, including consent withdrawal, please see our Privacy Policy

Reviewer #1: No

Reviewer #2: **Yes: ** Nasim Shakouri

---

## [Author Response · Author response to Decision Letter 1]

26 Jun 2025

Please find the response in the attachment "Response to Reviewers.docx"

---

## [Decision Letter · Decision Letter 1]

Static load test and bearing capacity analysis of broken line pretensioned prestressed concrete I-beam

PONE-D-25-21465R1

Dear Dr. Chenning Song,

We’re pleased to inform you that your manuscript has been judged scientifically suitable for publication and will be formally accepted for publication once it meets all outstanding technical requirements.

Kind regards,

Jeyashree T M, Ph. D

Academic Editor

PLOS ONE

Additional Editor Comments (optional):

Reviewers' comments:

Reviewer's Responses to Questions

**Comments to the Author**

Reviewer #1: All comments have been addressed

Reviewer #2: All comments have been addressed

2. Is the manuscript technically sound, and do the data support the conclusions?

Reviewer #1: Yes

Reviewer #2: Yes

3. Has the statistical analysis been performed appropriately and rigorously?

Reviewer #1: Yes

Reviewer #2: Yes

4. Have the authors made all data underlying the findings in their manuscript fully available?

Reviewer #1: Yes

Reviewer #2: Yes

5. Is the manuscript presented in an intelligible fashion and written in standard English?

Reviewer #1: Yes

Reviewer #2: Yes

Reviewer #1: Thanks for the revision. I think the revision has addressed my comments from previous round. No further questions.

Reviewer #2: I am pleased to recommend it for acceptance. The authors have effectively addressed all the previous comments, enhancing the clarity and impact of their work.

**Do you want your identity to be public for this peer review?** For information about this choice, including consent withdrawal, please see our Privacy Policy

Reviewer #1: No

Reviewer #2: **Yes: ** Nasim Shakouri
